

# Comparative analysis of the microRNA transcriptome between yak and cattle provides insight into high-altitude adaptation

Jiuqiang Guan[1,*], Keren Long[2,*], Jideng Ma[2,*], Jinwei Zhang[2], Dafang He[2], Long Jin[2], Qianzi Tang[2], Anan Jiang[2], Xun Wang[2], Yaodong Hu[2], Shilin Tian[1,3], Zhi Jiang[3], Mingzhou Li[2] and Xiaolin Luo[1]

[1] Yak Research Institute, Sichuan Academy of Grassland Science, Chengdu, China
[2] Institute of Animal Genetics and Breeding, College of Animal Science and Technology, Sichuan Agricultural University, Wen'jiang, China
[3] Novogene Bioinformatics Institute, Beijing, China
[*] These authors contributed equally to this work.

## ABSTRACT

Extensive and in-depth investigations of high-altitude adaptation have been carried out at the level of morphology, anatomy, physiology and genomics, but few investigations focused on the roles of microRNA (miRNA) in high-altitude adaptation. We examined the differences in the miRNA transcriptomes of two representative hypoxia-sensitive tissues (heart and lung) between yak and cattle, two closely related species that live in high and low altitudes, respectively. In this study, we identified a total of 808 mature miRNAs, which corresponded to 715 pre-miRNAs in the two species. The further analysis revealed that both tissues showed relatively high correlation coefficient between yak and cattle, but a greater differentiation was present in lung than heart between the two species. In addition, miRNAs with significantly differentiated patterns of expression in two tissues exhibited co-operation effect in high altitude adaptation based on miRNA family and cluster. Functional analysis revealed that differentially expressed miRNAs were enriched in hypoxia-related pathways, such as the HIF-1α signaling pathway, the insulin signaling pathway, the PI3K-Akt signaling pathway, nucleotide excision repair, cell cycle, apoptosis and fatty acid metabolism, which indicated the important roles of miRNAs in high altitude adaptation. These results suggested the diverse degrees of miRNA transcriptome variation in different tissues between yak and cattle, and suggested extensive roles of miRNAs in high altitude adaptation.

# INTRODUCTION

The yak is a unique livestock species that lives in the Qinghai-Tibet Plateau and adjacent areas, at altitudes of 2,500–6,000 m (*Nivsarkar et al., 1997*). Yak and cattle is a pair of closely related species and diverged five million years ago (*Qiu et al., 2012*). Although yak and cattle are different species, their genomes have strong similarities, including an identical number

Corresponding authors
Mingzhou Li,
mingzhou.li@sicau.edu.cn
Xiaolin Luo, luoxl2004@sina.com

of chromosomes (30 chromosomes), similar karyotypes (*Wiener, Jianlin & Ruijun, 2003*) and the extensive synteny which cover up to 94% of the yak genome (*Qiu et al., 2012*), leading to the availability of these two species in deciphering high altitude adaptation mechanism. Yak have evolved some anatomical and physiological traits that equip them for the extreme high-altitude environment, including larger lungs and hearts (*Wiener, Jianlin & Ruijun, 2003*), blunted hypoxic pulmonary vasoconstriction (*Dolt et al., 2007*), a strong environmental sense (*Wiener, Jianlin & Ruijun, 2003*) and a high energy metabolism (*Wang et al., 2011*). The mechanism of high altitude adaptation has become a topic of great interests in recent years. Extensive and in-depth investigations of high-altitude adaptation have been carried out at the level of morphology (*Hoppeler et al., 1990*), anatomy (*Ahmad et al., 2016*), hemodynamics (*Ahmad et al., 2016*), physiology (*Monge & León-Velarde, 1991*) and genomics (*Huertasánchez et al., 2013*; *Qiu et al., 2012*), but few investigations focused on the role of miRNA in high-altitude adaptation.

microRNAs (miRNAs) are a kind of endogenous noncoding RNA which post-transcriptionally modulate gene expression through either translational repression or mRNA degradation (*Bushati & Cohen, 2007*). Recent studies identified a large amount of hypoxia-induced miRNAs that widely participate in various biological processes, including cell apoptosis (*Lynam-Lennon, Maher & Reynolds, 2009*), DNA repair (*Hu & Gatti, 2011*; *Landau & Slack, 2011*), cell cycle (*Chen & Hu, 2012*), angiogenesis (*Nicoli et al., 2010*) and metabolism regulation (*Kulshreshtha et al., 2007*). For example, myocardial hypoxia induces expression of the miR-199a:214 cluster which thereafter downregulates several cardiac and mitochondrial targets, such as *PPARδ*, downregulation of *PPARδ* provokes glycolytic metabolism in heart failure (*El et al., 2013*).

We investigate: (a) whether miRNAs was involved in the mechanism of high altitude adaptation, (b) the kind of high altitude-adaptation miRNAs, (c) the functions of these high altitude-adaptation miRNAs. We constructed 11 small RNA cDNA libraries of heart and lung tissues in three female yaks and three female cattle, and performed small RNA sequencing. In total, we identified 757 and 769 unique mature miRNAs in yak and cattle, respectively. The further analysis revealed stronger expression divergence in lung than heart between yak and cattle. In two tissues, we identified total of 85 differentially expressed (DE) miRNAs between yak and cattle, which showed potential co-operation effect in high altitude adaptation. The functional enrichment analysis revealed extensive roles of miRNAs in hypoxia-related pathways, such as the HIF-1α signaling pathway, the insulin signaling pathway, the PI3K-Akt signaling pathway, nucleotide excision repair, cell cycle, apoptosis and fatty acid metabolism.

## MATERIALS AND METHODS

### Ethics statement

All animals were conducted according to the Institutional Animal Care and Use Committee in College of Animal Science and Technology, Sichuan Agricultural University, Sichuan, China under permit No. DKY-B20151608. Yak and cattle were collected from A'ba (altitude, 3,000 m) and Chengdu (altitude, 485 m), respectively. Animals were feed with free access to food and water, and were killed humanely.

## Sample collection

Three unrelated 2-year old adult females for both of yaks and cattle (Luxi Huang cattle) were used in this study. Two of significant hypoxia-responsive tissues (heart and lung) were rapidly collected from each carcass, washed three times with physiological saline, immediately frozen in liquid nitrogen. All frozen samples were stored at −80 °C until RNA extraction.

## RNA isolation, library preparation, and sequencing

The total RNA were extracted with Trizol (Ambion, USA). NanoDrop ND-2000 spectrophotometer (Nano Drop, Wilmington, DE, USA) and Bioanalyzer 2100 (Agilent Technologies, Santa Clara, CA, USA) were used to monitor the concentration and integrity of RNA, respectively. In brief, several successive steps consist the Illumina sequencing. The small RNA with length of 14–40 nt were first purified by polyacrylamide gel electrophoresis (PAGE), and then specific adapters were ligated to the purified small RNA. The ligated RNA were reverse transcribed to cDNA libraries. Finally, each library were sequenced on Genome Analyzer.

## Data analysis of small RNA sequencing

The analysis procedure of miRNA identification and quantification was conducted as previous study (*Li et al., 2010*) with several modifications. The raw reads were processed using Illumina's Genome Analyzer Pipeline software. After trimming off the adapters, the resulting reads was successively filtered by some strict criterial (only reads with size ranging from 14 to 27 nt, containing no more than 80% A, C, G or T; containing no more than two N (undetermined bases) were retained). Bovine known classes of RNAs (i.e., mRNA, rRNA, tRNA, snRNA, snoRNA and repeats) were subsequently removed through searching against three databases, including NCBI (*Pruitt et al., 2009*), Rfam (*Gardner et al., 2008*) and Repbase (*Lukasz et al., 2006*). All retained reads were defined as "high quality reads". Then, "high quality reads" were mapped to bovine genome and mapped reads were retained. All retained "high quality reads" were mapped to the known bovine pre-miRNAs and known pre-miRNAs from 24 other mammals in miRBase 20.0 (*Kozomara & Griffithsjones, 2014*); reads mapping to bovine or other mammalian miRNAs were defined as miRNA candidates. Given the reliability of miRNA identification, miRNA candidates with a read count ≥3 at least one sample were retained for subsequently analysis. Comparable numbers of miRNA species and similar expressional patterns were identified for using bovine genome or yak genome (Fig. S1), we selected bovine genome as the reference genome in this study.

## Identification of differentially expressed miRNAs

To reveal the difference of miRNA transcriptome between yak and cattle, we identified the differentially expressed (DE) miRNAs using edgeR package (*Robinson, Mccarthy & Smyth, 2010*). To obtain the normalized expression levels, read counts were loaded into edgeR and normalized using the supplied trimmed-mean-of-M-values algorithm. DE miRNAs are defined as those miRNAs with a folder change >2 and a Benjamini Hochberg FDR (False Discovery Rate) <0.05 between yak and cattle.

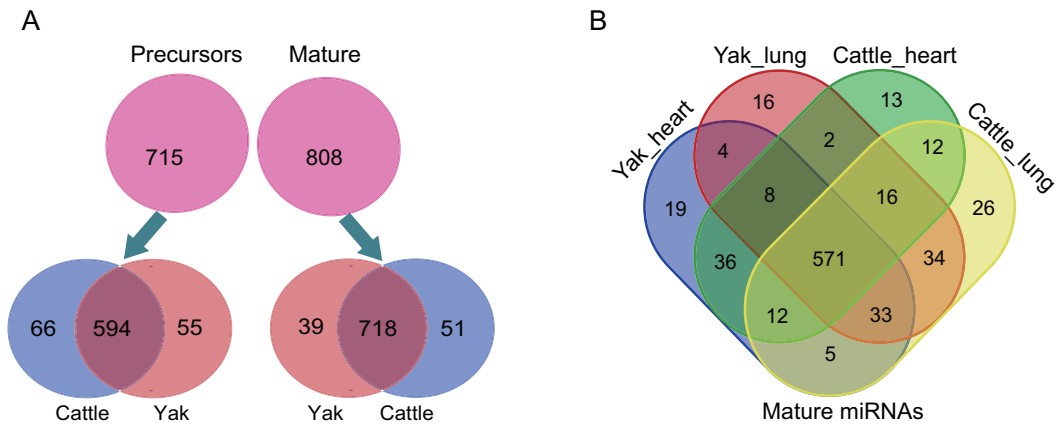

**Figure 1** **The overview of small RNA sequencing.** (A) Mature miRNAs and miRNA precursors identified in this study. (B) Venn charts indicate expression patterns of mature miRNAs among each sample.

## The prediction and functional analysis of miRNA target genes

As a miRNA pathway analysis web-server, DIANA-mirPath (*Vlachos et al., 2015*) is able to identify miRNA target, and molecular Gene Ontology (GO) and pathways (http://www.microrna.gr/miRPathv3). The predictions in this study were based on the database of human beings, owing to the lack of the bovine miRNAs in mirPath3.0.

## RESULTS

### Overview of small-RNA sequencing

We constructed and sequenced 11 small RNA libraries, including five heart libraries (i.e., two yak hearts and three cattle hearts) and six lung libraries (i.e., three yak lungs and three cattle lungs). A total of 96.19 million (M) raw reads were obtained from the 11 libraries, with an average of 8.74 M ($8.74 \pm 3.37$ M) in each library. Through a series of strict criteria, a total of 84.57 M raw reads were defined as high quality reads; With an average of $7.69 \pm 2.68$ M raw reads in each library were defined as "high-quality reads" (Table S1). By aligning these data to the Rfam database, we also identified abundant known non-coding RNAs (e.g., tRNAs, rRNAs, snoRNAs and snRNAs) (Fig. S1). Consistent with the canonical size of Dicer-processed products, over 80% of high-quality reads were 21–23 nt in length ($81.03 \pm 1.72\%$, $n = 11$), and the highest percentage of high-quality reads were 23 nt in length ($45.17 \pm 1.62\%$, $n = 11$), followed by 23 nt ($19.09 \pm 2.25\%$, $n = 11$) and 21 nt ($16.68 \pm 1.65\%$, $n = 11$) (Fig. S1). We then examined small RNA profiling of heart and lung for cattle and yak. In total, we identified 808 mature miRNAs, corresponding to 715 pre-miRNAs (Fig. 1A). Majority of these mature miRNAs were widely expressed in all sample and only a few were expressed as tissue or species-specific (Fig. 1B). The biological replicates showed high Spearman's correlation coefficient (average $r = 0.99$), which indicates strong experimental confidence (Fig. S3).

## Different organs showed varying degrees of miRNA expression divergence

To obtain an overview of miRNA transcription patterns, we performed a clustering analysis of the expression of 718 co-expressed miRNAs. As expected, all samples were firstly clustered by tissue and thereafter by species (Fig. 2A), which indicates the longer evolutionary history of these organs than the species. In addition, we also conducted principal component analysis. The first eigenvector distinguished tissue differentiation, while the second eigenvector captured the biological differentiation and high altitude-related differentiation between yak and cattle for heart and lung tissues. Lung showed larger differentiation than heart between yak and cattle (Fig. 2B). This result is consistent with the universal organ-specific pattern of mRNA gene expression (*Brawand & Al, 2011*; *Melé et al., 2015*). It was previously reported that heart showed stronger mRNA transcriptome variance than lung between yak and cattle (*Wang et al., 2015*), but the miRNA transcriptome showed the opposite trend in this study, which indicates miRNA and mRNA transcriptome inconsistence. Although organ divergence contributes to the primary variance, heart and lung showed similar expression profiles of the most abundant miRNAs (Fig. 2C). Overall, the two organs shares five miRNAs of the top 10 expressed miRNA, which included bta-miR-143-3p, bta-miR-22-3p, bta-miR-27b-3p, bta-miR-30e-5p and bta-miR-30a-5p.

## Differentially expressed miRNAs showed a potential co-operation effect in high-altitude adaptation

To further reveal the miRNA adaptation of yak to high altitudes, we performed differential analysis for lung and yak between yak and cattle. In total, we detected 85 differentially expressed mature miRNAs between yak and cattle (fold change > 2, FDR < 0.05), of which 29 were present in heart (Fig. 3A) and 70 in lung (Fig. 3B). The detailed information is listed in Table 1 and Tables S2, S3. The greater amount of DE lung miRNAs is consistent with the lower Spearman's correlation coefficient for lung, increases the reliability of greater miRNA transcriptome divergence of lung than heart between yak and cattle.

It is well known that miRNAs that derive from a common ancestor can be grouped into a family, and family members often have similar biological function. Of these 85 differentially expressed miRNAs, we detected six miRNA families, including miR-2285, miR-34, miR-192, miR-449, miR-200 families (Table S4). We also detected eight miRNA clusters in differentially expressed miRNAs that included from two members to 13 members, such as the mir-215/206, mir-127/136 cluster (Table S5). The largest cluster was mir-154/376/379 cluster, which consisted of 13 members. To explore the expression pattern of miRNA families and clusters, we calculated the Pearson's correlation coefficient of each family, cluster and background. The background showed a wide Pearson's correlation range from $-0.99$ to $0.99$, with an average of $0.1221$ ($r = 0.1221 \pm 0.55$, $n = 257,403$). Compared with background, the miRNA families and clusters showed high levels of Pearson's correlation, with coefficients of 0.86 in miRNA clusters ($r = 0.86 \pm 0.23$, $n = 478$) and 0.44 ($r = 0.45 \pm 0.53$, $n = 52$) in miRNA families (Fig. 3C), which indicated the co-expression and potential similar functions of members in each miRNA family or cluster. These results are consistent with the observation that genes which cluster together share common
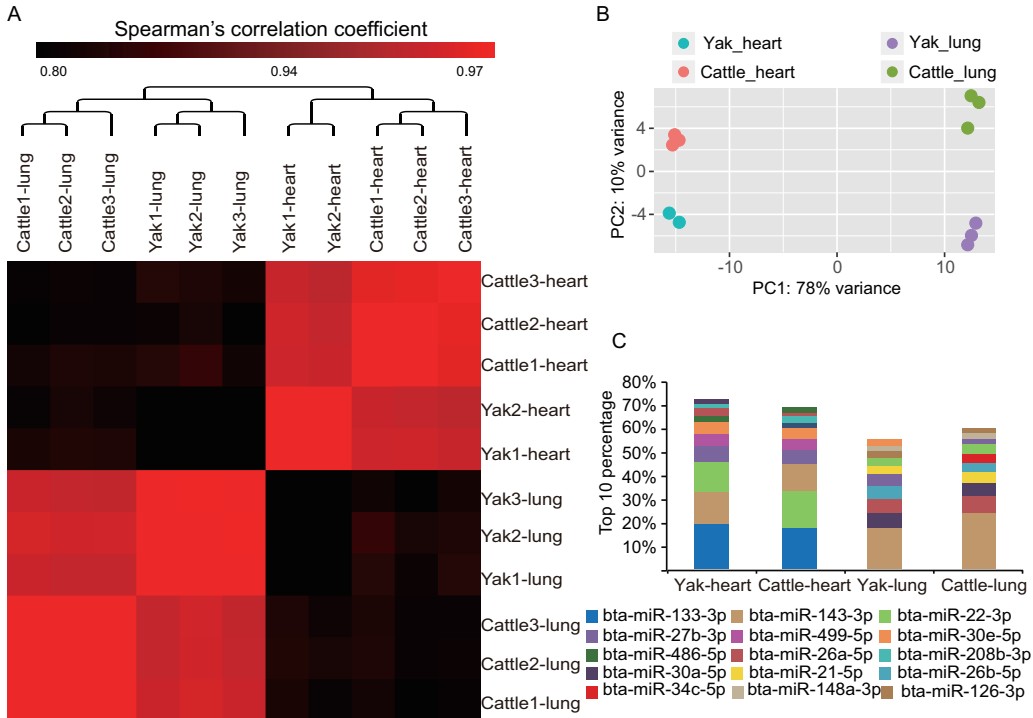

**Figure 2   Highly conserved heart and lung tissues showed varying degrees of divergence.** (A) Hierarchical clustering and heat map matrix of pairwise Spearman's correlations of the counts of 718 co-expressed miRNAs between 11 miRNA libraries. (B) Principal component analysis (PCA) plot of each sample. The fraction of the variance explained is 78% for eigenvector 1 and 10% for eigenvector 2. (C) Top 10 unique miRNAs with the highest expression levels in heart and lung tissues of yak and cattle.

functions and similar expression pattern (*Eisen et al., 1998*). Consistent with the hypoxic high-altitude environment, these miRNA families and clusters showed high relevance to high altitude adaptation. For example, the p53-responsive microRNA miR-192 and miR-215 are able to induce cell cycle arrest (*Braun et al., 2008*) and significantly inhibited cellular nucleotide excision repair (NER) (*Xie et al., 2011*). We identified two differentially expressed miRNAs not only belonging to the same family but also belonging to the same cluster. These miRNAs included miR-34b/34c/449a/449b, miR-200a/429. The miR-34 family consist of four members, including miR-34a, miR-34b, miR-449a, miR-449b. miR-34a/b family clustered together, and miR-449a/b family clustered together (Figs. 3D, 3E), and showed similar expression pattern between yak and cattle. miRNAs in each family showed overall similar expression pattern in heart and lung tissues between yak and cattle, indicating the potential similar effects of these miRNAs in high-altitude adaptation. The miR-34 family members showed hypoxia related functions, overexpression of miR-34a is able to regulate the G1/S checkpoint through targeting several genes, such as CDK6 and E2F, et al. (*Chen & Hu, 2012*; *Sun et al., 2008*), and miR-34c affects the S-phase checkpoint through binding to c-myc (*Cannell & Bushell, 2010*). In brief, the similar expression pattern of miRNA families and clusters suggested the potential collaboration effect of miRNAs in high altitude adaptation.

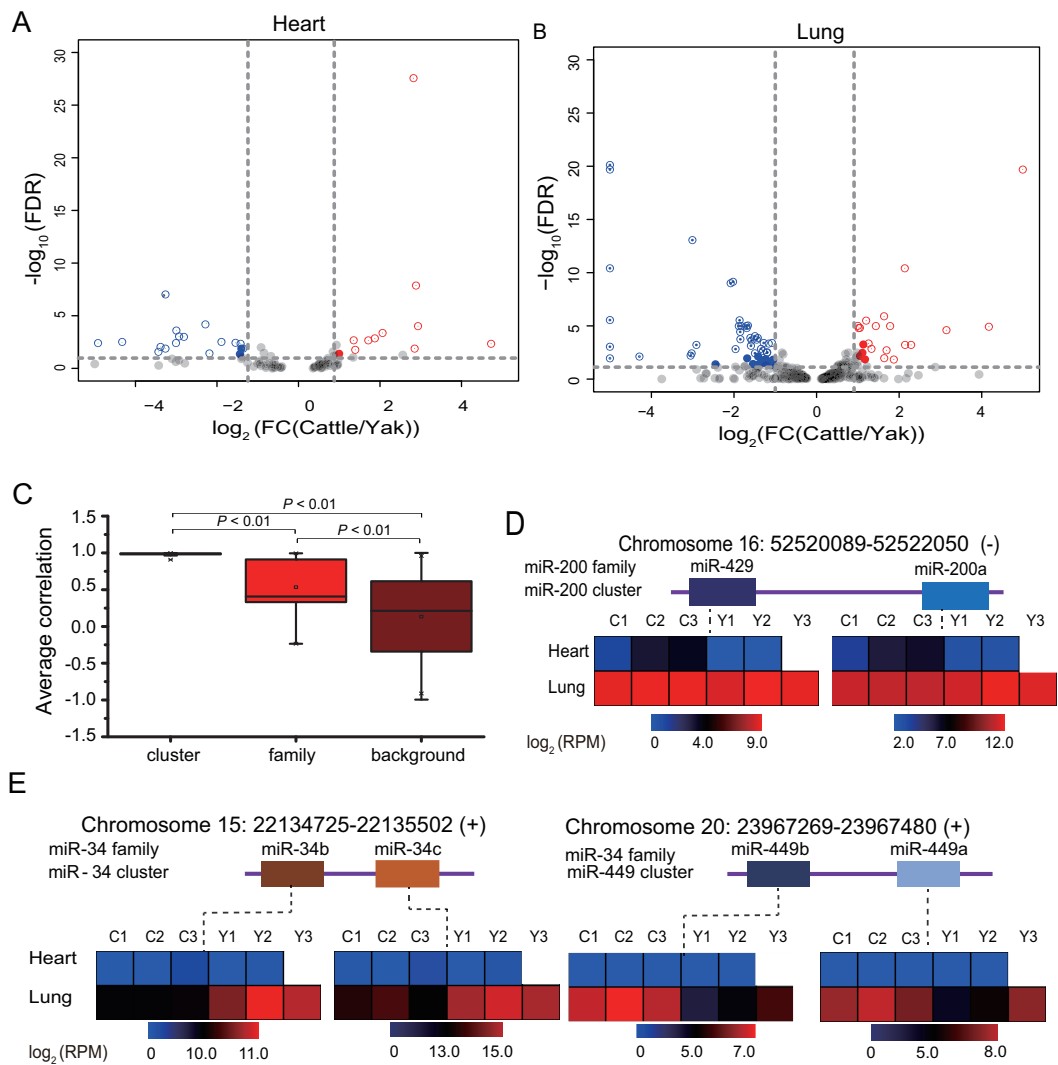

**Figure 3** **Expression pattern of differentially expressed miRNAs belong to the miRNA family in sequence and miRNA cluster in location.** (A–B) Volcano plot showing differentially expressed miRNAs in heart (A) and lung (B) between yak and cattle. (C) Box plot of Pearson's correlation coefficient for miRNA families, clusters and background. (D) Expression pattern of miR-200 family belonging to miR-200 cluster. (E) Expression pattern of miR-34 family belonging to miR-34 cluster and miR-449 cluster.

**Table 1** **Differentially expressed miRNAs in heart and lung tissues between yak and cattle.**

| Categories | Number of differentially expressed miRNAs | | |
|---|---|---|---|
| | Total | Upregulated in yak | Upregulated in cattle |
| Heart | 29 | 11 | 18 |
| Lung | 70 | 22 | 48 |

## DE miRNAs are associated with hypoxia-related functions

To illustrate the potential function of identified DE miRNAs, we therefore performed target prediction and functional enrichment analysis by DIANA online software (*Vlachos et al., 2015*). As expected, the majority of DE miRNAs were involved in hypoxia-related pathways, such as the HIF-1α signaling pathway, the insulin signaling pathway, the PI3K-Akt signaling pathway, nucleotide excision repair, cell cycle, apoptosis, fatty acid metabolism, which were high relevant to high-altitude adaptation (Figs. 4A, 4B). In particular, the insulin signaling pathway, fatty acid biosynthesis, the p53 signaling pathway and cell cycle were simultaneously enriched in both tissues, which indicates the central role of these DE miRNAs in high-altitude adaptation pathways.

As a sequence-specific transcription factor, p53 regulates the expression of several genes associated with growth arrest (*Kracikova et al., 2013*), DNA repair, apoptosis, and angiogenesis (*Farhang et al., 2013*). In this study, we identified 31 DE miRNAs that enriched in P53 signaling pathway, of which 14 was presented in heart and 22 in lung (Table 2). Especially, five miRNAs showed differential expression in both heart and lung, including miR-192-5p, miR-32-3p, miR-200b-3p, miR-146b-5p, miR-122-5p. We interestingly found that miR-192, miR-32, miR-122, miR-200b are regulators of p53 feedback circuit (Fig. 4C). For example, miR-32 is in a recurrent autoregulatory circuit in which p53 regulate the expression of miR-32, and on other hand, miR-32 indirectly increase p53 accumulation through targeting Mdm2 and TSC1 (*Suh et al., 2012*). In addition, p53 can posttranscriptionally regulated the expression of miR-122, and in turn miR-122 can suppress p53 activity by activating Akt (*Manfè et al., 2012*), thus constructing a negative feedback. Three (miR-192-5p, miR-32-3p, miR-122-5p) of these five miRNAs showed decreased expression level in yak, and one miRNA (miR-125-5p) was upregulated in yak. The differential expression of these p53 feedback circuit regulators may play an important roles in the p53 accumulation.

It is well known that some intrinsic and extrinsic genotoxic stress such as oxidative stress and ultraviolet light could result in DNA damage (*Jackson & Bartek, 2009*). Because of the hypoxia and intensive ultraviolet environment at high altitudes, yak could have evolved strong abilities to resist DNA damage caused by hypoxia and intensive ultraviolet. By conducting a literature search, we identified 10 known DNA damage response miRNAs, including miR-504-5p, miR-34a, miR-215, miR-192, miR-128, miR-181, et al (Fig. S4A). These miRNAs are involved in many facets of DNA damage response, including DNA repair, the cell cycle checkpoint and apoptosis. It was reported that miR-181a and miR-181b were upregulated in aggressive breast cancers, and their expression level inversely correlates with ataxia telangiectasia mutated (ATM) levels and DNA damage repair (*Bisso et al., 2013*). As a negative regulator of human p53, miR-504 is able to directly bind to two target sites in the p53 3′-untranslated region, and play important roles in the process of p53-mediated cell-cycle arrest and apoptosis (*Hu et al., 2010*). The functional enrichment analysis by DIANA software indicated that target genes of 28 DE miRNAs in lung were also enriched in nucleotide excision repair, which indicates a substantial involvement of DE miRNAs in DNA damage response (Fig. S4B).

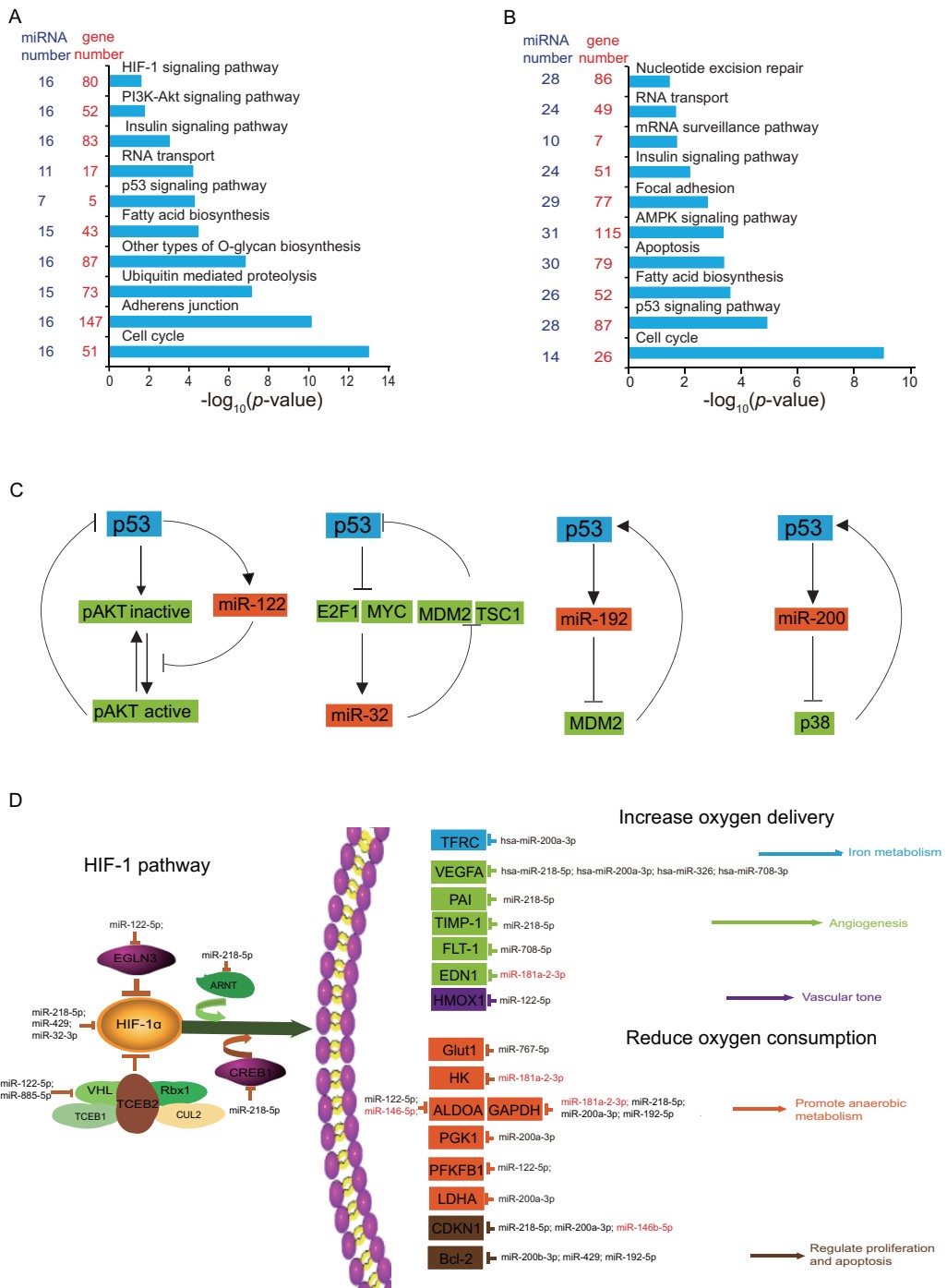

**Figure 4 Functional analysis of DE miRNAs in heart and lung.** (A–B) Gene Ontology (GO) categories and pathways enriched for target genes of DE miRNAs in heart (A) and lung (B). The *p* value indicating the significance of the enrichment, was corrected by Fisher's exact test. miRNA number was the number of DE miRNAs involved in each GO term. Gene number was the number of target genes for DE miRNAs involved in each GO term. (C) DE miRNAs involved in p53 feedback circuit. (D) DE miRNAs and corresponding target genes are involved in HIF-1 pathway. miRNAs in red indicated upregulated in yak, miRNAs in black indicated downregulated in yak.

**Table 2 DE miRNAs enriched in P53 signaling pathway.**

| Categories | miRNA name | Heart | Lung | Target genes in Tarbase |
|---|---|---|---|---|
| DE in two tissues | miR-192-5p | down | down | CDKN2,MDM2, EI24, PERP, ZMAT3 |
| | miR-32-3p | down | down | MDM4, CDK4, ZMAT3, CCNG1, PPM1D, CASP3 |
| | miR-200b-3p | down | up | MDM4, CCNE1, CDK2, PMAIP1, ZMAT3, SIAH1, DDB2, RCHY1 |
| | miR-146b-5p | up | up | ATR, CDKN1A, THBS1, CCNB3 |
| | miR-122-5p | down | down | MDM4, ATM, TP53, CCNE1, GTSE1, CD262, BCL2L4, CCNG1, TP73, CYCS |
| DE in heart | miR-200a-3p | down | no | CDKN1A, CCNE1, SIAH1, THBS1, RRM2B |
| | miR-708-5p | down | no | SESN3, CCNB3 |
| | miR-429 | down | no | CDK2, PMAIP1, SIAH1 |
| | miR-885-5p | down | no | CDK2 |
| | miR-194-5p | down | no | TSC2 |
| | miR-708-3p | down | no | CASP3 |
| | miR-208a-3p | down | no | CHK1 |
| | miR-181a-2-3p | down | no | MDM2, CDK4, GTSE1, BCL2L4, SERPINE1 |
| | miR-218-5p | down | no | ATR, ATM, TP53, CDKN1A, CDK4, BCL2L4, ZMAT3, IGFBP3, SERPINE1, SESN3, CCND3, CD262, RCHY1 |
| DE in lung | miR-424-5p | no | up | IGFBP3, RRM2B, SESN3, IGFBP3, CCNG1, SIAH1, MDM2, CHK1, CDKN1A |
| | | | | CDK4, GTSE1, RFWD2, CD262, PERP, ZMAT3, PPM1D, SCOTIN |
| | miR-582-5p | no | up | ATR, CDK4, CCND3, PERP, SIAH1, PTEN, CCNG1, SIAH1, CYCS, CCNB3 |
| | miR-34b-5p | no | up | CDKN1A, CDK4, CD262, SIAH1, PPM1D, THBS1 |
| | miR-450b-5p | no | up | MDM2, CDK4, THBS1, SESN3 |
| | miR-34c-5p | no | up | CDK4, STEAP3, SERPINE1 |
| | miR-136-5p | no | up | THBS1 |
| | miR-582-3p | no | up | CDK4, PTEN, CCNB3 |
| | miR-19b-3p | no | up | MDM2,MDM4,ATM,CHK1,TP53,CDKN1A |
| | miR-379-5p | no | down | SERPINE1, CASP3 |
| | miR-382-5p | no | down | PTEN, TSC2, CASP3 |
| | miR-181c-3p | no | down | MDM2, EI24 |
| | miR-299-5p | no | down | CDKN1A, SERPINE1 |
| | miR-449b-5p | no | down | CDK4, SERPINE1, THBS1, STEAP3 |
| | miR-551b-3p | no | down | ZMAT3 |
| | miR-127-3p | no | down | THBS1 |
| | miR-127-5p | no | down | THBS2 |
| | miR-122-3p | no | down | CD262 |

**Notes.**
"Down" indicated downregulated in yak; "up" indicated upregulated in yak, "no" indicated not differently expressed in particular tissue.
HIF-1α is a basic transcription factor that trans-activates genes which encode proteins that participate in homeostatic responses to hypoxia, it include expression of proteins involving in glucose metabolism (*Remels et al., 2015*), cell proliferation (*Scaringi et al., 2013*), and vascularization (*Skuli et al., 2012*; *Zhao et al., 2012*). In this study, we detected 15 HIF-1 α signaling related miRNAs in heart, and these miRNAs can regulate both upstream and downstream genes of the HIF-1 α signaling pathway (Fig. 4D). It was reported that miR-429 construct a negative regulatory loop with HIF-1 and decrease HIF-1 activity through targeting HIF1A mRNA before HIF-2 is activated (*Bartoszewska et al., 2015*). MiR-326, a recognized tumor-suppressing miRNA, has been found to target the 3′-UTR of the human CCND1 mRNA and promote cell apoptosis (*Sun et al., 2016*). Hypoxia-induced microRNA-424 promoted angiogenesis both *in vitro* and *in vivo*, and increased the expression levels of HIF-1 α and HIF-2 α by targeting cullin 2 (CUL2), a key protein for the assembly of the ubiquitin ligase system (*Ghosh et al., 2010*). In addition, miR-218 repression increases the abundance and activity of multiple RTK effectors, and promotes the activation of hypoxia-inducible factor (HIF), most notably HIF2 α (*Mathew et al., 2014*).

Some metabolism-related DE miRNAs and pathways were also revealed in this study. For example, the DNA damage causing miR-181a inhibit the expression of genes that were associated with lipid synthesis and increase the expression of genes that were associated with β-oxidation. It is able to cause decreased expression of isocitrate dehydrogenase 1 which plays roles in tricarboxylic acid (TCA) cycle, thereafter inhibits lipid accumulation (*Chu et al., 2015*). These results are consistent with the energy shift from fatty acid oxidation to glucose oxidation and glycolysis in the high-altitude living conditions (*Ding et al., 2014*; *Ge et al., 2012*).

## DISCUSSION

In recent years, the mechanism of high altitude adaptation have become topic of great interests and attracted many specialists from various areas. They carried out extensive and in-depth investigations of high altitude adaptation at the levels of morphology (*Hoppeler et al., 1990*), physiology (*Monge & León-Velarde, 1991*) and genomics (*Huertasánchez et al., 2013*; *Qiu et al., 2012*) in human and other species. Yak and cattle is a pair of closely related species and diverged five million years ago (*Qiu et al., 2012*), and provide an ideal model for deciphering the mechanism of high altitude adaptation for their high similarity of genome. Here, we performed transcriptome analysis in yak and cattle for two hypoxia-sensitive tissues, heart and lung. We identified comparable numbers of miRNA species using yak and cattle genome, respectively. In addition, we also detected similar expressional patterns across all libraries using yak and cattle genome, with the same heat map matrix of pairwise Spearman's correlations.

In this work, we identified 808 mature miRNAs, corresponding to 715 pre-miRNAs, covering majority of bovine miRNA in miRBase 20.0. The Spearman's correlation analysis

revealed that lung showed larger differentiation than heart between yak and cattle at the level of miRNA transcriptome, which is inconsistent with previous study that heart showed stronger variance than lung between yak and cattle at mRNA transcriptome (*Wang et al., 2015*). We identified a large amount of differentially expressed miRNA between yak and cattle, including 29 miRNAs in heart and 70 miRNAs in lung. Further examination revealed that majority of differentially expressed miRNAs are downregulated in yak, including 18 of 29 DE miRNAs in heart and 48 of 70 DE miRNAs in lung (Table 1). The pervasive downregulation of differentially expressed miRNAs in yak may correspond to the upregulation of target pathways. For example, we detected 15 differentially expressed miRNAs enriched in HIF1 α signaling pathway in heart (Fig. 4D), and 13 miRNAs were downregulated in yak which may result in the overall upregulation of HIF1 α signaling pathway, consistent with previous studies that HIF pathway genes played important roles in high altitude adaptation and underwent positive selection in high altitude populations in humans (*Beall et al., 2010*; *Simonson et al., 2010*) and other species (*Li et al., 2010*; *Qiu et al., 2012*). In addition, we also identified 14 miRNAs were differentially expressed genes in heart and enriched in the p53 signaling pathway, of which 13 miRNAs are downregulated in heart. It was reported that alleles of ubiquitin specific peptidase 7 (USP7), interleukin 6 family cytokine (LIF), and murine double minute2 (MDM2) were selected for in relation to harsh environmental variables related to high altitude (*Jacovas et al., 2015*). The widespread of differentially expressed miRNAs targeting the p53 pathway gene indicated the potential involvement of p53 pathway in high altitude adaptation. Taken together, these results indicated the potential roles of these DE miRNAs in yak high altitude adaptation through upregulating important pathways.

Although we found substantial differentially expressed miRNA that may be associated with high altitude adaptation, some limitations in this study should be noted. Since diverging five million years ago, yak and cattle underwent distinct selective pressure resulting in inherent genetic difference which may to some extent disturb the result and explanation. In addition, human miRNA orthologs were used to perform miRNA target prediction which may suffer from the miRNA-mRNA interaction bias in different species. A more comprehensive association study of mRNA and miRNA expression is necessary in order to better reveal the authentic roles of differentially expressed miRNAs.

## CONCLUSIONS

In this study, we illustrated the differences in the miRNA transcriptomes for heart and lung between yak and cattle, and suggested extensive roles of miRNAs in high altitude adaptation. The work performed here will provide a typical demonstration for future deciphering of the mechanism of high altitude adaptation.

## ACKNOWLEDGEMENTS

We thank YuJie Wang, Wanling Qiu, Zihu Hu and Can Liu for help with experiments.

### Funding

This work was supported by grants from the National Natural Science Foundation of China (31601918, 31522055 and 31401073), the National Program for Support of Top-notch Young Professionals, the Program for Innovative Research Team of Sichuan Province (2015TD0012), the Fund of Fok Ying-Tung Education Foundation (141117) and the Key Project of Sichuan Education Department (15ZA0003, 16ZA0025 and 15ZA0008). The funders had no role in study design, data collection and analysis, decision to publish, or preparation of the manuscript.

### Grant Disclosures

The following grant information was disclosed by the authors:
National Natural Science Foundation of China: 31601918, 31522055, 31401073.
Program for Innovative Research Team of Sichuan Province: 2015TD0012.
Fok Ying-Tung Education Foundation: 141117.
Sichuan Education Department: 15ZA0003, 16ZA0025, 15ZA0008.

### Competing Interests

Shilin Tian and Zhi Jiang are employees of Novogene Bioinformatics Institute.

### Author Contributions

- Jiuqiang Guan and Keren Long conceived and designed the experiments, analyzed the data, wrote the paper.
- Jideng Ma conceived and designed the experiments, analyzed the data, prepared figures and/or tables.
- Jinwei Zhang, Dafang He, Long Jin, Anan Jiang, Xun Wang, Yaodong Hu performed the experiments.
- Qianzi Tang analyzed the data, contributed reagents/materials/analysis tools.
- Shilin Tian and Zhi Jiang analyzed the data.
- Mingzhou Li and Xiaolin Luo conceived and designed the experiments, contributed reagents/materials/analysis tools, reviewed drafts of the paper.

### Animal Ethics

The following information was supplied relating to ethical approvals (i.e., approving body and any reference numbers):

This study was approved by the Institutional Animal Care and Use Committee in College of Animal Science and Technology, Sichuan Agricultural University, Sichuan, China.

### Data Availability

The small RNA sequencing reads have been deposited into the NCBI gene expression omnibus (GEO) under the accession number GSE87833.

## Supplemental Information

Supplemental information for this article can be found online at http://dx.doi.org/10.7717/peerj.3959#supplemental-information.

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
