# Peer review of "Comparative analysis of the microRNA transcriptome between yak and cattle provides insight into high-altitude adaptation"

_PeerJ, doi:10.7717/peerj.3959_

## Round 0.1 · original submission · Minor Revisions

The topic fits the field of the journal. The study is well designed and performed. Please revise the manuscript based on the reviewers comments and re-submit it.

Reviewer 1 ·

Basic reporting

In general, the manuscript is well written. However, the following issues should be addressed.

The growing environment of the experiment animals should be provided, especially the altitude. The altitude could potentially affect the overall result. In the section of differential expression of microRNA, more detail on how to normalize read counts should be provided. The method of FDR calculation is also needed.

In Result section,
Line 120, it is hard to interpret (88.86±4.87%).
Lline 131, it is not traditional way to use correlation coefficient to measure experimental confidence, what is the method to calculate the correlation coefficient: Pearson or Spearman’s?
Line 166~169, how the Pearson’s correlation of family, cluster and background are calculated is not clear, between yak and cattle or between tissues? If it is between yak and cattle, how are tissues treated in the calculation. More details are needed to interpret the result.

Figure1. ‘coserved’ should be ‘conserved’
Figure 2. The color scale for the heatmap is too wide.
Figure 3. Legend does not match the figure. There are 5 panels in the figure, while there are only A to C in legend.
Figure 4. p-value should use –log10 instead of –log2. Fold change in p value does not make sense and is hard to interpret.
Table S1: Format the spreadsheet to show all numbers.
Table S2: Round or format the numbers to an easily reading format.
Table S5: Does ‘genomeID’ represent chromosome?
All the supplementary tables need to be reformatted to a readable and scientifically meaningful format.
All supplementary figures should have figure legends.

Experimental design

no comment

Validity of the findings

no comment

Reviewer 2 ·

Basic reporting

This manuscript presents the findings from miRNA profiling study of Yak and Cattle. It is well written with a professional article structure. The findings sound interesting to relevant research community.

Experimental design

No comments

Validity of the findings

Some minor comments on the data analysis

In the method section of data analysis for small RNA sequencing, the author wrote "Given the reliability of analysis results, reads with a copy number larger than three were retained for subsequently analysis. Comparable amount of miRNAs were identified for using bovine genome or yak genome (data not shown), we selected bovine genome as the reference genome in this study." More details need to be added here. For example, does the "copy number" mean "read count"? need to be greater than three in all samples or at least one sample? Regarding miRNAs identified with bovine and yak genome, the number of miRNAs might be similar, how about their expression level? If the expression level is not similar, the results from yak genome will have to be used.

In the section for identifying miRNA targets, how are the ortholgous human miRNAs identified for bovin miRNAs, by comparing sequence similarity or simply by matching the names?

Reviewer 3 ·

Basic reporting

Revision by a native English speaking colleague or editing services is required.

The introduction would benefit from some discussion of the evolutionary relationship between yak and cattle. Related to this point, the introduction should provide some justification for using cattle as the comparison species in this study.

The legend for Figure 4 requires additional details for panel D.

Figure 4 panel D contains text that has been resized such that the aspect ratio is incorrect.

The legend for Figure 4 includes the statement "miRNAs in red indicated upregulated in heart, miRNAs in black indicated downregulated in yak". This color scheme appears inadequate as the two possibilities are not mutually exclusive.

The data from this study is available through NCBI GEO under the accession number GSE87833. The GEO record for this study contains errors that should be corrected. For example, the methods section contains information from a different study: "Retinal mRNA profiles of 21-day-old wild-type (WT) and neural retina leucine zipper knockout (Nrl−/−) mice". The GEO summary also includes the word "epigenetic": "we illustrated the epigenetic differences in the microRNA transcriptomes level for heart and lung between yak and cattle". It isn't clear to me why the term "epigenetic" is used in this context--the authors have not demonstrated that these expression differences arise from epigenetic mechanisms as opposed to DNA sequence differences.

Experimental design

The sample size is small but appears suitable for interspecies comparisons. However, more information on the animals is needed. Specifically, were they maintained at high or low elevations?

The section "Data analysis of small RNA sequencing" contains some confusing and inconsistent statements. First, the statement "Given the reliability of analysis
results, reads with a copy number larger than three were retained for subsequently analysis" is unclear. What is the "analysis results" referring to? Also, what does it mean for a read to have "a copy number larger than three"? Does this refer instead to a miRNA with support from more than three reads? Second, the statement "Comparable amount of miRNAs were identified for using bovine genome or yak genome (data not shown), we selected bovine genome as the reference genome in this study." The section up to this point makes no mention of using a reference genome--instead reads are mapped to bovine and other mammalian pre-miRNAs. Thus the meaning of this statement is not clear.

The section "Data analysis of small RNA sequencing" refers to a database as "NCBI". A more specific name for this database is needed.

Validity of the findings

In general the results are interesting but do not, in my mind, convincingly indicate that the miRNA expression differences between cattle and yak relate to high altitude adaptation. A more convincing study would have incorporated mRNA expression, such that the differences in miRNA expression could be linked to target gene expression differences. In general the language used to describe the conclusions needs to be adjusted to better convey the indirect nature of the findings. For example, the conclusion "revealed extensive roles of miRNAs in high altitude adaptation" could be "suggest extensive roles of miRNAs in high altitude adaptation."

The paper does not include a discussion section. A discussion section should be added and should include a discussion of the limitations of this work, including the limitations of miRNA target prediction.

The discussions of the implications of the miRNA expression differences do not seem to note which miRNAs are expressed more highly in yak and which are expressed more highly in cattle. Figure 4 does not clearly convey this information. This information should be included this in the discussions and in figure 4. For the miRNA differential expression results to support the hypothesis that miRNA expression is involved in high altitude adaptation it is important that the direction of the expression differences be discussed.

---

## Round 0.2 · Minor Revisions

Please clarify the comments raised by Reviewer 3 and re-submit the revised manuscript.

Reviewer 1 ·

Basic reporting

The issues raised in last review have been taken care of. No further comment

Experimental design

The experimental design meets the journal standard. No further comment.

Validity of the findings

No comment

Reviewer 3 ·

Basic reporting

Basic reporting
* * *
1. The authors indicate that an editing service has been used. However, the writing does not seem to have improved in quality and thus still requires extensive editing and is not suitable for publication in its current form.

2. The figures still contain text shown at incorrect aspect ratios.

3. The legend for Figure 4 still requires information describing panel D.

4. "reads mapping to bovine or other mammalian miRNAs were defined as conserved miRNAs": based on this statement, reads mapping only to bovine would be classified as "conserved", which doesn't make sense.

5. "reads with a read count larger than three" is unclear. I assume the authors mean "miRNAs supported by three or more reads".

6. line 122: "Given the reliability of analysis results"--In the revised version it is still not clear to me what this statement refers to.

7. line 124" "Comparable amount of miRNAs" should be "Comparable numbers of miRNA species"

8. lines 130 and 131: the normalization methods and FDR methods should be stated rather than just referring, for example, to a "default" method.

9. the discussion includes results that are not presented in the results section, for example "we identified 784 and 807 miRNAs using yak and cattle genome, respectively". The discussion should not present new results.

Experimental design

Experimental design:
* * *
10. Based on the miRBase information on microRNA nomenclature, orthology should not be inferred by name alone:

http://www.mirbase.org/help/nomenclature.shtml

"The numbering of miRNA genes is simply sequential. For instance, at the time of writing the last published miRNA was mouse mir-352. The next novel published miRNA will get the number 353. However, if you submit an Xenopus miRNA that is identical to human mir-121 for example, we will suggest you also name your sequence mir-121."

"Please note that miRNA names are able to convey only limited information, and are entirely unsuitable to encode information about complex sequence relationships. You should not therefore rely on the name to tell you all you need to know about the sequence. Sensible database approaches should instead use dedicated fields and annotation to describe such relationships, such as the "family" data provided here."

Based on the author's response to reviewer 2 and on the methods of the paper, the authors have not used a suitable, sequence-based approach to confirm that they have identified the appropriate human ortholog miRNAs for inferring mRNA targets. This raises questions about the validity of all the downstream functional analysis. The authors need to apply a more rigorous ortholog detection procedure, and they need to clearly describe this procedure in the methods.

Validity of the findings

I have concerns about the validity of the findings given the issue with the miRNA ortholog identification described under Experimental design.

---

## Round 0.3 · accepted · Accept

The study was well designed and performed. The topic fits in the scope of this journal. The authors made extensive revision based on the reviewers' comments. I believe the paper is ready for publication in the journal. Congratulations!